# Estimating Nutrient Uptake Requirements for Melon Based on the QUEFTS Model

**Meijuan Wen [1], Sicun Yang [1,\*], Lin Huo [1], Ping He [2] , Xinpeng Xu [2] , Chengbao Wang [1], Yueqiang Zhang [3] and Wei Zhou [2]**

1 Institute of Soil, Fertilizer and Water-Saving Agriculture, Gansu Academy of Agriculture Sciences, Lanzhou 730070, China; wen_mj@126.com (M.W.); gshuolin@163.com (L.H.); wcb17901@163.com (C.W.)
2 Institute of Agricultural Resources and Regional Planning, Chinese Academy of Agricultural Sciences, Beijing 100081, China; heping02@caas.cn (P.H.); xuxinpeng@caas.cn (X.X.); zhouwei02@caas.cn (W.Z.)
3 Interdisciplinary Research Center for Agriculture Green Development in Yangtze River Basin, College of Resources and Environment, Southwest University, Chongqing 400716, China; zhangyq82@swu.edu.cn
\* Correspondence: yangsicun@sina.com

**Abstract:** Imbalanced and excessive fertilizer application has resulted in low yields and reduced nutrient use efficiency for melon production in China. Estimating nutrient requirements is crucial for effectively developing site-specific fertilizer recommendations for increasing yield and profit while reducing negative environmental impacts. Relationships between the yield and nutrient uptake requirements of above-ground dry matter were assessed using 1127 on-farm observations (2000–2020) from melon production regions of China. The quantitative evaluation of fertility of tropical soils (QUEFTS) model was used to estimate nutrient requirements. It predicted a linear increase in yield at balanced nutrient uptake levels until the yield reached approximately 60–80% of the potential yield. In order to produce 1000 kg of fruit, 2.9, 0.4 and 3.2 kg/ha of N, P and K (7.2:1.0:7.8), respectively, were required for above-ground parts, while the corresponding nutrient internal efficiencies were 345.3, 2612.6 and 310.0 kg per kg N, P and K, respectively, whereas 1.4, 0.2 and 1.9 kg of N, P and K were required to replace nutrients removed after harvest. The corresponding fruit absorption rates were 47.0%, 59.5% and 58.2%, respectively. Field validation experiments confirmed the consistency between observed and simulated uptake rates, indicating that this model could estimate nutrient requirements. These findings will help develop fertilizer recommendations for improving melon yield and nutrient use efficiency.

**Keywords:** QUEFTS model; melon; nutrient requirements; internal efficiency (IE)

## 1. Introduction

Melon (*Cucumis, melo* L.) is an important horticultural crop that is highly appreciated worldwide owing to its high nutritive and medicinal value. The main melon-producing country is China, followed by Turkey, Iran and India (1.8 to 1.2 million tons per year) [1]. According to the 2017 China Agriculture Yearbook, melon with a harvested area of 14 million ha and yield of 45 million ton at the end of 2017 is an important export crop of China [2]. The main melon-producing regions are located in the semi-arid regions of northwestern China. These regions supply a large proportion of melons for the internal market. With the development of high-yielding varieties, factors such as fertilizer application, cultivation method and farmland irrigation strategy play important roles in achieving high yield and improving fruit quality characteristics [3]. In particular, appropriate nutrient input is important. In Poland, under drip irrigation conditions, N, P and K fertilizers for the highest melon yields were 120 kg/ha, 100 kg/ha and 150 kg/ha [4]. However, according to a recent fertilization survey, N application rates for melon in northern China reached more than 400–600 N kg/ha [5]; meanwhile, more than 50% of areas received high P and K inputs, with an unreasonable nutrient ratio [6]. Excessive and imbalanced application of

chemical fertilizers has resulted in both reduced melon yield and inefficient fertilizer use, while exacerbating various environmental issues, including the release of greenhouse gases, eutrophication and soil quality degradation [7], and it is detrimental to the production of high-quality melons. Therefore, knowing precise nutrient requirements is critical in soil and crop management for optimizing agronomic performance and improving fruit quality, as well as for maintaining agricultural sustainability by reducing environmental risks.

The estimation of nutrient uptake requirements is necessary for nutrient management and fertilizer recommendations. Previous studies on fertilizer recommendations for melon production in China focused mainly on soil testing and plant tissue nutrient uptake. Fertilizer recommendations frequently based on soil testing and target yield have been reported to increase melon yield and nutrient use efficiency [8]. However, these methods were mostly based on individual or few test points, with high heterogeneity. Moreover, it is expensive and time consuming to collect multi-year and multi-location soil samples in a smallholder farming system. Plant-based fertilizer recommendations, such as above-ground canopy nutrition diagnosis, have been applied for rice nutrient management [9]. In this method, the estimation of crop nutrient uptake is required to balance crop removal in order to achieve a specific target yield [10]. Furthermore, although previous research was carried out at research stations, not much attention was paid to the quantitative uptake of plant nutrients [11]. Individualized fertilizer recommendations for different regions or crops cannot be based on a single nutrient absorption parameter, as this may limit the production potential of high-yield varieties and the improvement of fertilizer utilization.

Site-specific nutrient management (SSNM) is an alternative approach for dynamic management of nutrients to optimize the supply and demand of nutrients within a specific field in a particular cropping season; this strategy is practiced in China, India, Indonesia, the Philippines and West Africa [12]. Its use has been reported to increase yield and nutrient use efficiency while lowering environmental risks [13]. Thus, SSNM-advocated fertilizer recommendations could be used to determine quantitative crop nutrient requirements. However, the wide variation in agronomic practices, nutrient supply and climate condition at the time of planting affects the requirement and uptake of nutrients in melon production, thereby making it difficult to extrapolate results from experimental stations to small fields. Therefore, attention must be paid to establish fertilization recommendations for melon nutrient management based on the scientific analysis of Big Data and empirical models that encompass a wide range of conditions.

The quantitative evaluation of the fertility of tropical soil (QUEFTS) model is used to estimate plant nutrient uptake requirements for a given yield level [14]. This model was selected for the present study because the interactions of N, P and K are taken into consideration in this model. Additionally, as this model is based on large-scale nutrient uptake data, errors in estimation that may occur when the results of only a small number of research stations are taken into consideration for guiding fertilization can be avoided. This model quantifies nutrient requirements under different target yields and predicts the relationship between crop yield and nutrient uptake of above-ground plant dry matter at maturity, following a function a linear and parabolic plateau pattern [15]. Thus, the QUEFTS model can be used to develop a quantitative understanding of crop nutrient requirements that is suitable for practical nutrient management under dynamic environmental conditions [16,17]. This model can provide a generic approach for estimating nutrient requirements for crops under a given target yield, which can be applied for site-specific nutrient management [10,18,19].

Presently, the QUEFTS model is being successfully applied in different countries including China for various crops such as maize [20,21], wheat [22], tea [23], rice [24], potato [25], radish [26] and watermelon [27]. However, it has not yet been used to estimate the nutrient requirements of melons in China, as little information on nutrient management and fertilizer recommendation for melon is available. The present study aimed at the following: (1) determine the relationship between melon yield and nutrient uptake across a wide range of melon-producing environments in China; (2) quantify the balanced

requirements of N, P and K for melon using the QUEFTS model; and (3) evaluate the accuracy of the nutrient uptake requirements of melon estimated using the QUEFTS model by conducting on-farm experiments in China.

## 2. Materials and Methods

### 2.1. Data Source

For this study, data for melon yield; N, P and K uptake by the fruit, stem and leaf; harvest index (HI); and soil properties were generated by field experiments conducted by our group during 2017–2020, as well as from unpublished field experiments conducted by the International Plant Nutrition Institute (IPNI) China Program and published articles dating from 2009 to 2017. Data from Knowledge Resource Integrated Database (https://kns.cnki.net (accessed on 31 May 2020)) from 2000 to 2020, covering 14 provinces or municipalities and representing different cultivation environments with variable soil agronomic practices and climatic conditions, were used in this study. A total of 1127 field experiments were considered. Specific fertilization practices such as farmer practices, optimal nutrient practices, nutrient omission treatments, long-term field experiments, different fertilizer application rates under variable nutrient management practices and cultivation practices, including irrigation and the control of pests and diseases, were in line with local management practices for ensuring high yields. Table 1 shows the soil characteristics.

**Table 1.** Soil characteristics in main melon-producing regions of China.

| Province | Cases (n) | Longitude (°E) | Latitude (°N) | pH | Organic Matter (%) | Alkali-Hydrolysable N (mg/kg) | Olsen P (mg/kg) | NH$_4$OAc-K (mg/kg) |
|---|---|---|---|---|---|---|---|---|
| Shandong | 24 | 112.59–117.77 | 34.65–40.16 | 7.5–8.0 | 1.8–2.0 | 74.3–98.8 | 18.7–27.2 | 76.1–211.2 |
| Jiangsu | 14 | 117.24–120.89 | 31.22–34.55 | 7.6–8.1 | 2.0–3.3 | 53.6–186.8 | 12.4–21.3 | 76.8–98.5 |
| Shanghai | 40 | 121.46–122.16 | 31.11–31.23 | 6.3–8.0 | 1.6–2.7 | 80.2–198.0 | 38.5–96.7 | 89.6–110.5 |
| Guangdong | 10 | 116.23–116.54 | 23.78–23.89 | 4.6–6.3 | 1.9–5.1 | 78.5–110.6 | 10.4–66.3 | 71.3–150.3 |
| Hainan | 6 | 109.36 | 19.20 | 6.63 | 2.0 | 98.8 | 27.2 | 211.0 |
| Hubei | 24 | 109.24–114.22 | 30.29–32.65 | 5.6–7.6 | 1.5–4.2 | 68.5–152.2 | 15.6–39.8 | 85.2–123.0 |
| Henan | 2 | 115.63 | 33.56 | 7.81 | 1.7 | 19.2 | 10.9 | 139.6 |
| Hebei | 6 | 115.26 | 36.13 | 7.4 | 1.16 | 32.0 | 45.0 | 50.0 |
| Inner Mongolia | 70 | 107.50–107.52 | 41.12–41.35 | 8.2–8.5 | 0.8–1.2 | 61.0–97.4 | 14.5–24.3 | 134.0–198.0 |
| Tianjin | 119 | 112.59–115.93 | 34.48–41.60 | 7.8–8.1 | 2.3–4.2 | 110.3–150.3 | 35.3–63.5 | 105.3–186.0 |
| Ningxia | 68 | 105.15–105.18 | 36.15–36.20 | 8.2–8.7 | 0.6–1.1 | 24.0–119.6 | 16.3–116.5 | 19.7–252.3 |
| Xinjiang | 94 | 76.45–76.49 | 39.12–39.14 | 8.1–8.5 | 1.1–2.6 | 12.3–56.2 | 8.4–22.2 | 120.1–208.3 |
| Shanxi | 51 | 108.25–108.30 | 34.20–35.25 | 7.5–8.2 | 9.3–15.4 | 68.7–96.4 | 10.9–41.9 | 130.2–177.5 |
| Gansu | 463 | 103.02–104.55 | 36.53–40.29 | 8.1–8.9 | 9.5–16.7 | 86.3–100.1 | 8.23–74.4 | 106.9–216.7 |

### 2.2. Development of the QUEFTS Model

The QUEFTS model describes the relationship between melon yield and nutrient absorption of the total above-ground dry matter from a large amount of data using a linear and parabolic plateau function [28]. Nutrient internal efficiency (IE, melon yield per unit of nutrient uptake in the above-ground parts) is a core parameter, representing the ability of the crop to convert nutrients obtained from various sources into economic yield [29]. By using the QUEFTS model, IE was assumed as a constant until the aimed yield reached 60–70% of potential yield. The precondition was to estimate required parameters *a* (minimum nutrient internal efficiency), *d* (maximum nutrient internal efficiency) and the yield potential. In order to ensure sensitivity of parameters a and d in the model, the upper and lower 2.5th, 5.0th and 7.5th percentiles of all calculated IE data were excluded. The potential yield was set as the maximum available yield under specific experimental conditions. Additionally, we used a Solver model in Microsoft Office Excel to estimate nutrient uptakes under different potential yields (20–60 t/ha) and the target yield (60 t/ha) and used this information to estimate the balanced nutrient uptake. The key steps of the QUEFTS model, which are described in detail in previous studies [10,16,30,31], are as follows: (a) defining the maximum accumulation and the maximum dilution borderlines of N, P and K and the yield potential to be achieved; (b) using the Solver model in Microsoft

Office Excel to estimate nutrient uptake under different potential yields; and (c) simulating optimal N, P, and K uptake curves under different target yields or potential yields.

*2.3. Field Validation*

Between 2018 and 2020, 39 farm field experiments were conducted in under different climatic conditions and soil types by using different varieties of melon across the northwest regions of China, including Ningxia (4 fields), Inner Mongolia (10 fields), Gansu (19 fields) and Xinjiang (6 fields), to analyze the relationship between the measured nutrient absorption of above-ground matter of melon and nutrient absorption estimated by the QUEFTS model. These regions are mostly deserts with continental climates and semi-arid conditions. Rainfall during the growing season is approximately 62 mm, 81 mm, 88 mm and 49 mm in Ningxia, Inner Mongolia, Gansu and Xinjiang, respectively. Melon is a popular local fruit, typically cultivated in dry soil under irrigated conditions in open fields in late April or early May and harvested in mid-July or late July. The Nutrient Expert system (NE), based on the modified SSNM and QUEFTS model and developed by IPNI, is a simple deliverable computer software application that is used to determine optimal nutrient requirements [10,32]. The feasibility of this system and the effect of fertilizer recommendations on melon cultivation have been verified [5]. Therefore, nutrient recommendations were based on the NE system for melon. Recommendations for N application were determined by yield response and agronomic efficiency, while P and K fertilizer recommendations were determined by yield response and nutrient removal rate under a certain target yield. Nutrient removal amount was determined by simulating optimal nutrient absorption using the QUEFTS model. The fertilizer recommendations for N, P and K were determined mainly by considering the crop planting system and residual effects of previously applied fertilizers to avoid the excessive addition of nutrients to the soil [33]. Yield response was calculated as the difference in yield gap between the plots that received ample N, P and K and the omission plots in which one of the nutrients was omitted. Agronomic efficiency of N, P and K was represented by an increment in yield per unit of applied N, $P_2O_5$ and $K_2O$. In accordance with NE recommendations, N, P and K fertilizers were applied as shown in Table 2. N was applied as urea (46% N), while P and K were applied as calcium superphosphate (18% $P_2O_5$) and potassium sulfate (50% $K_2O$), respectively. The best management practices were followed throughout the growth period of melon to eliminate suboptimal growth. The total irrigation volume during the growth period was 300 mm, with a total of seven irrigations carried out after mulching to ensure adequate bottom moisture content. The irrigations were carried out at intervals of 15 days after sowing. The cultivation method followed was plastic mulch-furrow irrigation.

**Table 2.** Rates of fertilizer application for nutrient expert recommendations.

| Province | Cases (n) | Fertilizer Application Rate (kg/ha) | | |
| --- | --- | --- | --- | --- |
| | | N | $P_2O_5$ | $K_2O$ |
| Inner Mongolia | 10 | 300 | 106 | 191 |
| Gansu | 19 | 225–309 | 88–93 | 83–200 |
| Xinjiang | 6 | 317 | 119 | 214 |
| Ningxia | 4 | 225 | 93 | 88 |

Three replications were carried out for all experiments. The planting pattern followed was 40 cm × 230 cm, 130 cm × 46 cm, 80 cm × 120 cm and 80 cm × 220 cm in Ningxia, Inner Mongolia, Gansu and Xinjiang, respectively. Other field management practices were performed based on the best local management practices. P fertilizer was applied in a single dose as basal fertilizer, and N fertilizer application was split into three doses, a base application (mid-April) and followed by two top dressing in late May and mid-June in the ratio 4:3:3, while K fertilizer application was split into two dose applications and a top dressing in the ratio 2:3. Additional field management practices including

irrigation, and weed and pest control were followed according to the optimum local management strategies.

After harvesting at maturity, plant samples, including the stems, leaves and fruits, were collected and oven dried at 80 °C for the determination of dry matter weight. Subsamples were digested using $H_2SO_4$-$H_2O_2$, and N, P and K concentrations were measured using the Kjeldahl method and Vanadomolybdate yellow color method [34], and atomic adsorption spectrophotometer (SpectAA-50/55, Varian, Australia), respectively. The total nutrient accumulation of N, P and K was calculated by multiplying the values of the above-ground dry matter with those of N, P and K concentrations. The obtained values were evaluated by using the QUEFTS model to determine the correlation between simulated and observed nutrient uptakes.

The root mean square error (RMSE) and normalized RMSE (n RMSE) were used to evaluate the feasibility of the QUEFTS model and the deviation between measured and simulated nutrient data [31] as follows:

$$RMSE = \sqrt{\frac{\sum_{i=1}^{n}\left(s_{i-m_i}\right)^2}{n}}$$
$$\text{n-RMSE} = \frac{RMSE}{\overline{m}}$$

where $s_i$ and $m_i$ represent simulated and measured nutrient uptake values (kg/ha), respectively, n represents the number of data and $\overline{m}$ represents the mean value of the measured nutrient uptake value (kg/ha). The equation for n-RMSE was applied to validate the accuracy of the model.

### 2.4. Statistical Analysis

SAS software (V8, SAS Institute Inc., Cary, NC, USA) was used to analyze the significance of differences between the mean values of simulated and measured nutrient uptake based at the 5% significance level.

## 3. Results and Discussion

### 3.1. Fleshy Fruit Yield and Nutrient Uptake

The average fresh fruit yield of melon (90.5% moisture content) in China during the period from 2000 to 2020 was 36.2 t/ha and ranged from 3.7 to 83.7 t/ha. The average yield was greater than the average yield in other parts of the world at 27.3 t/ha [1] and also higher than the 34.6 t/ha reported in the 2017 China Agriculture Statistical Report [2]. These differences were mainly due to the diversity of melon varieties and the differences in nutrient management practices and cultivation technology adopted. Additionally, melon cultivation is more adaptable to areas with high light intensity and temperatures compared to that of other crops. Regions such as Gansu, Ningxia and Inner Mongolia follow scientific and standardized management practices for the selection of varieties, cultivation of seedlings, transfer cultivation and harvesting of fruits. Furthermore, many farmers have been given professional training to achieve high-quality and yield [6,35,36]. The high melon yield in the present study was probably due to the adoption of best field management practices and the use of balanced N, P and K fertilization along with the best agronomic measures, while lower yields may be the result of omission experiments. The average dry matter weight of stems and leaves was 3121.8 t/ha and ranged from 744.1 to 10,842.4 t/ha. The value HI was 0.58, 0.65 and 0.59 for N, P and K, respectively, revealing that 58%, 65% and 59% of N, P and K of above-ground plant parts were stored in the fruit (Table 3). Accordingly, the quantities of N, P and K in the fruits harvested from the field were used to assess fertilizer N, P and K replacement requirements in order to achieve the target yield and maintain levels of P and K in the soil.

**Table 3.** Statistics of melon yield and stem and leaf yield; nutrient uptake in fruit; nutrient uptake in stem and leaf; N, P and K concentrations in fruit and stem and leaf; nutrient uptake in above-ground dry matter; and N, P and K harvest index (HI).

| Parameter | Unit | N [1] | Mean | SD [2] | Minimum | 25% Q [3] | Median | 75% Q | Maximun |
|---|---|---|---|---|---|---|---|---|---|
| fresh melon fruit yield | t/ha | 1123 | 36.2 | 12.3 | 3.7 | 27.9 | 38.7 | 42.7 | 83.3 |
| stem and leaf yield | kg/ha | 919 | 3121.8 | 1418.3 | 744.1 | 2115.5 | 3025.9 | 3634.6 | 10,842.5 |
| seed and melon yield | kg/ha | 900 | 3651.6 | 979.0 | 551.1 | 2991.4 | 3692.7 | 4259.7 | 7427.7 |
| N in fruit | kg/ha | 564 | 63.8 | 22.3 | 10.0 | 47.4 | 65.1 | 75.3 | 180.9 |
| P in fruit | kg/ha | 555 | 9.51 | 3.6 | 2.1 | 7.5 | 9.3 | 11.0 | 35.9 |
| K in fruit | kg/ha | 551 | 66.7 | 25.8 | 25.4 | 53.2 | 60.7 | 70.8 | 280.8 |
| N in stem and leaf | kg/ha | 547 | 47.3 | 20.3 | 14.7 | 47.4 | 65.1 | 75.3 | 158.8 |
| P in stem and leaf | kg/ha | 551 | 5.3 | 3.3 | 1.3 | 3.6 | 4.5 | 5.8 | 36.3 |
| K in stem and leaf | kg/ha | 547 | 49.6 | 27.3 | 7.6 | 32.7 | 40.9 | 63.0 | 192.1 |
| $N_c$ in fruit | g/kg | 720 | 50.5 | 9.9 | 23.7 | 44.4 | 50.8 | 56.1 | 89.6 |
| $P_c$ in fruit | g/kg | 719 | 5.7 | 2.4 | 4.5 | 1.7 | 5.0 | 6.2 | 16.9 |
| $K_c$ in fruit | g/kg | 719 | 51.5 | 33.5 | 13.8 | 33.6 | 39.4 | 61.9 | 244.1 |
| $N_c$ in stem and leaf | g/kg | 741 | 49.3 | 10.4 | 23.2 | 42.7 | 51.9 | 55.1 | 92.5 |
| $P_c$ in stem and leaf | g/kg | 714 | 9.7 | 3.1 | 3.3 | 8.1 | 10.0 | 10.9 | 30.71 |
| $K_c$ in stem and leaf | g/kg | 714 | 25.8 | 12.4 | 14.7 | 19.4 | 22.5 | 28.5 | 144.7 |
| N in total DM [4] | kg/ha | 623 | 117.9 | 44.4 | 42.2 | 93.7 | 111.3 | 127.5 | 174.2 |
| P in total DM | kg/ha | 573 | 14.7 | 4.3 | 5.6 | 14.4 | 16.5 | 19.1 | 33.3 |
| K in total DM | kg/ha | 572 | 120.9 | 53.1 | 38.6 | 86.7 | 104.0 | 145.8 | 391.3 |
| N Harvest index | g/kg | 557 | 0.58 | 0.12 | 0.19 | 0.56 | 0.62 | 0.65 | 0.85 |
| P Harvest index | g/kg | 547 | 0.65 | 0.11 | 0.31 | 0.63 | 0.69 | 0.72 | 0.87 |
| K Harvest index | g/kg | 547 | 0.59 | 0.09 | 0.23 | 0.57 | 0.60 | 0.63 | 0.88 |
| Harvest index (HI) | g/kg | 704 | 0.62 | 0.08 | 0.32 | 0.53 | 0.66 | 0.68 | 0.84 |

Note: n [1], number of observations. SD [2], standard deviation. Q [3], quartile. DM [4], dry matter, Nc, N concentration. Pc, P concentration. Kc, $_K$ concentration.

The nutrient concentrations within each plant component showed great variations (Table 3). The accumulation of N, P and K in the total dry matter was 117.9, 14.7 and 120.9 kg/ha and ranged from 44.4 to 174.3; 4.3 to 33.3; and 53.1 to 391.3 kg/ha, respectively. The average concentration of N, P and K in the fruit was 50.5, 5.7 and 51.5 g/kg, while the corresponding values in the stem and leaf were 49.3, 9.7 and 25.8 g/kg. The average accumulation in stem and leaf was 47.3, 5.3 and 49.6 kg/ha for N, P and K, which ranged from 14.7 to 158.8; 5.3 to 36.3; and 27.3 to 192.1 kg/ha, respectively. The accumulation of N, P and K in fruit was 63.8, 9.5 and 66.7 kg/ha, and ranged from 10.0 to 180.9 kg N/ha, 2.2 to 35.9 kg P/ha, and 25.4 to 280.8 kg K/ha, respectively (Table 3). We observed that the concentration of K stored in fruit was higher than that stored in the stem and leaf, which indicates that as a typical horticultural crop based on harvested fruits, melon production requires sufficient quantities of K to meet the demands of fruit growth and synthesis of metabolites [37]. The production of melon is similar to that of watermelon [27], which requires a higher N input than that required for grain crop [10,19]. These differences were mainly caused by variations in nutrient supply and changes in balanced nutrient uptake requirements as estimated by the QUEFTS model [11].

### 3.2. Internal Efficiency and Reciprocal Internal Efficiency

The QUEFTS model used internal efficiency (IE) and reciprocal internal efficiency (RIE) to estimate the relationship between melon yield and N, P and K nutrient absorption in the above-ground parts of the plant based on the transportation capacity of N, P, and K from the plant to fruit (Table 4). For the present study, IE and RIE datasets originated from several treatments, including optimal treatments, omission treatments and current farmer practices. The average IEs of N, P and K were 348.9, 2786.3 and 358.0 kg/kg, respectively, and ranged from 133.0 to 581.4 kg/kg for N; 349.0 to 6154.6 kg/kg for P; and 108.2 to 739.6 kg/kg for K. These results indicated that the variation range of IE for N, P, and K was broad, which might be due to the large variation within each species and site environment as well as in

other factors that affect nutrient absorption, including field management. Moreover, owing to immobile P resulting in lower nutrient availability, the IE of P was higher than that of N and K, which is different from that of grain crops such as maize and rice, but similar to that of vegetable crops such as potatoes and radish [25,26,38]. In order to produce a yield of 1000 kg of melon, the above-ground parts require an average of 2.9 kg of N, 0.4 kg of P and 3.1 kg of K ranging from 1.4 to 7.5 kg/1000 kg for N; 0.2 to1.3 kg/1000 kg for P; and 1.4 to 9.2 kg/1000 kg for K (Table 4). The amount of N and K required to produce 1000 kg of melon is approximately eight times the P requirement, because N and K are major nutrients for which their deficiencies can limit the growth and yield response of melon. Hu [39] reported that in order to produce 1000 kg of melon, 3.6–5.3 kg of N, 0.2–0.5 kg of P and 3.8–5.4 kg of K were required to reach a yield level of 40.2–71.1 t/ha., indicating that for a high yield greater amounts of N and K were required to produce 1000 kg of melon. The higher RIE of N and K was related to higher yield levels [19].

**Table 4.** Descriptive statistics of the internal efficiency (IE) of N, P and K (kilogram of fleshy fruit per kilogram of nutrient) and its reciprocal internal efficiency (RIE, kg nutrient per 1000 kg fleshy fruit) for melon grown in China.

| Parameter | Unit | N [1] | Mean | SD | Minimum | 25% Q [2] | Median | 75% Q | Maximun |
|-----------|------|-------|------|-----|---------|-----------|--------|-------|---------|
| IE-N | kg/kg | 759 | 359.0 | 73.3 | 133.0 | 322.1 | 364.7 | 401.6 | 632.7 |
| IE-P | kg/kg | 719 | 2725.4 | 722.1 | 349.0 | 2350.1 | 2690.9 | 3098.8 | 6154.6 |
| IE-K | kg/kg | 718 | 353.8 | 99.3 | 108.2 | 288.6 | 377.2 | 424.4 | 739.6 |
| RIE-N | kg/t | 760 | 2.9 | 0.8 | 1.4 | 2.5 | 2.7 | 3.1 | 7.5 |
| RIE-P | kg/t | 720 | 0.4 | 0.12 | 0.2 | 0.3 | 0.4 | 0.4 | 1.3 |
| RIE-K | kg /t | 719 | 3.1 | 1.2 | 1.4 | 2.4 | 2.7 | 3.4 | 9.2 |

[1] Number of observations; [2] 25% Q, Med., and 75% Q represent the 25th, 50th and 75th percentiles of the database, respectively.

### 3.3. Determining the Parameters to Adapt the QUEFTS Model

The values of parameters *a* and *d* of N, P and K were used to run the QUEFTS model, and calculations were performed by excluding the upper and lower 2.5th (Set I), 5.0th (Set II) and 7.5th (Set III) percentiles of nutrient IE of the combined data for all melon datasets (Table 5). The solver model in Excel was used for modeling the nutrient absorption rate of the three percentile sets. The nutrient requirements of the sets, calculated using the QUEFTS model, were similar, which only narrowed the distance between the maximum accumulation and maximum dilution, as observed in the curves of N, P and K requirements under the three sets of parameters, except when the potential yield was close to the target yield. Since Set I had a wide range of variability, it was selected to estimate balanced nutrient uptake and the relationship between melon yield and above-ground nutrient uptake (Figure 1). The values for constants *a* and *d* were 201, 1404 and 163 kg/kg and 510, 4183 and 511 kg/kg for N, P and K, respectively, in Set I. The borderline values of *a* and *d* observed in the present study were different from those reported in grain crops [10,30,40], which may be due to the fact that the model was calibrated based on dry matter; however, when it is calibrated on the basis of fresh matter, water content was also considered, and the results are in agreement with those for other crops. On the other hand, these differences in the borderline values between the maximum accumulation and dilution may be caused by variations in physiological characteristics and management conditions. The values of *a* and *d* for N and K were less than those for P, which indicated that, for the production of melon, higher quantities of N and K were required than P. The values of *a* and *d* in the present study indicated that N is the primary factor that determines melon yield, which is consistent with the conclusion of a previous study [41].

**Table 5.** Constants *a* and *d* were calculated by excluding the upper and lower 2.5th (Set I), 5.0th (Set II) and 7.5th (Set III) percentiles of all nutrient efficiency data of the combined dataset.

| Nutrients | Set I | | Set II | | Set III | |
|---|---|---|---|---|---|---|
| | a (2.5th) | d (97.5th) | a (5th) | d (95th) | a (7.5th) | d (92.5th) |
| N | 201 | 510 | 234 | 465 | 249 | 448 |
| P | 1404 | 4183 | 1486 | 3937 | 1644 | 3746 |
| K | 163 | 511 | 175 | 488 | 189 | 475 |

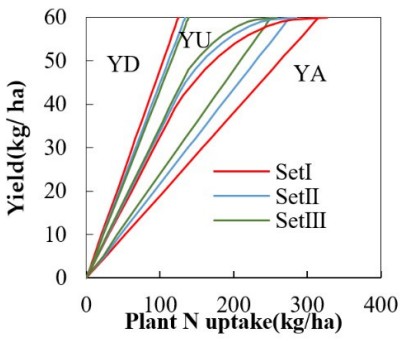 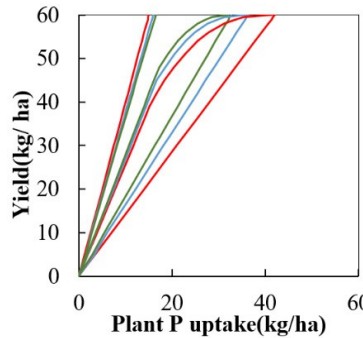 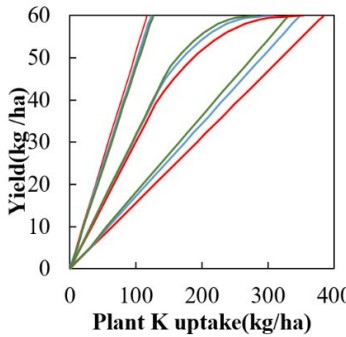

**Figure 1.** Relationship between melon yield and above ground nutrient absorption of N, P and K at three sets of constants a and d for all internal efficiency data (HI ≥ 0.4). YD, YA and YU represent maximum dilution, maximum accumulation and balanced uptake of N, P and K in plant dry matter, respectively. The potential yield was set at 60 t/ha.

### 3.4. Estimation of the Optimal Nutrient Uptake

The relationship between melon yield and above-ground nutrient absorption was estimated using the QUEFTS model under different potential yields ranging from 20 to 60 t/ha and using the *a* and *d* values of Set I in which the HIs below 0.4 were eliminated, because lower HI indicates disturbances in growth conditions caused by factors other than nutrients [16,28,42]. We set a target of 60 t/ha for the potential yield based on mean annual precipitation and the conditions of open field cultivation in China [43]. Accordingly, the highest target yield of 60 t/ha was set to run the QUEFTS model in order to estimate balanced nutrient requirements (Figure 2a–c). The model estimated the balanced requirements of N, P and K (YU) for different potential and target yields and predicted a linear increase in melon yield until the yield target reached approximately 60–70% of the potential yield. In other words, regardless of the differences in potential yield, the QUEFTS model predicted that the linear part of the response curve is always the same for the optimal nutrient accumulation of N, P and K to produce 1000 kg of fruit, reflecting that plant growth is mainly limited by nutrient supply (Figure 2a–c). As the target yield became closer to the potential yield, the parameterized QUEFTS model estimated an increase in nutrient uptake; however, a drastic decrease in IE and RIE was estimated when the target yield extended beyond the potential yield of 60–70%, where factors other than nutrient availability, such as soil moisture and occurrence of weeds, pests and diseases, also affected plant growth (Table 6). These observations were similar to those of previous studies on grain crops [38,40] and vegetable crops [25,44], and it is more likely that lower yield is associated with higher IE. Therefore, it is necessary to balance the application of N, P and K according to the levels of indigenous nutrients in soil or plant demand, instead of blindly pursuing a higher target yield.

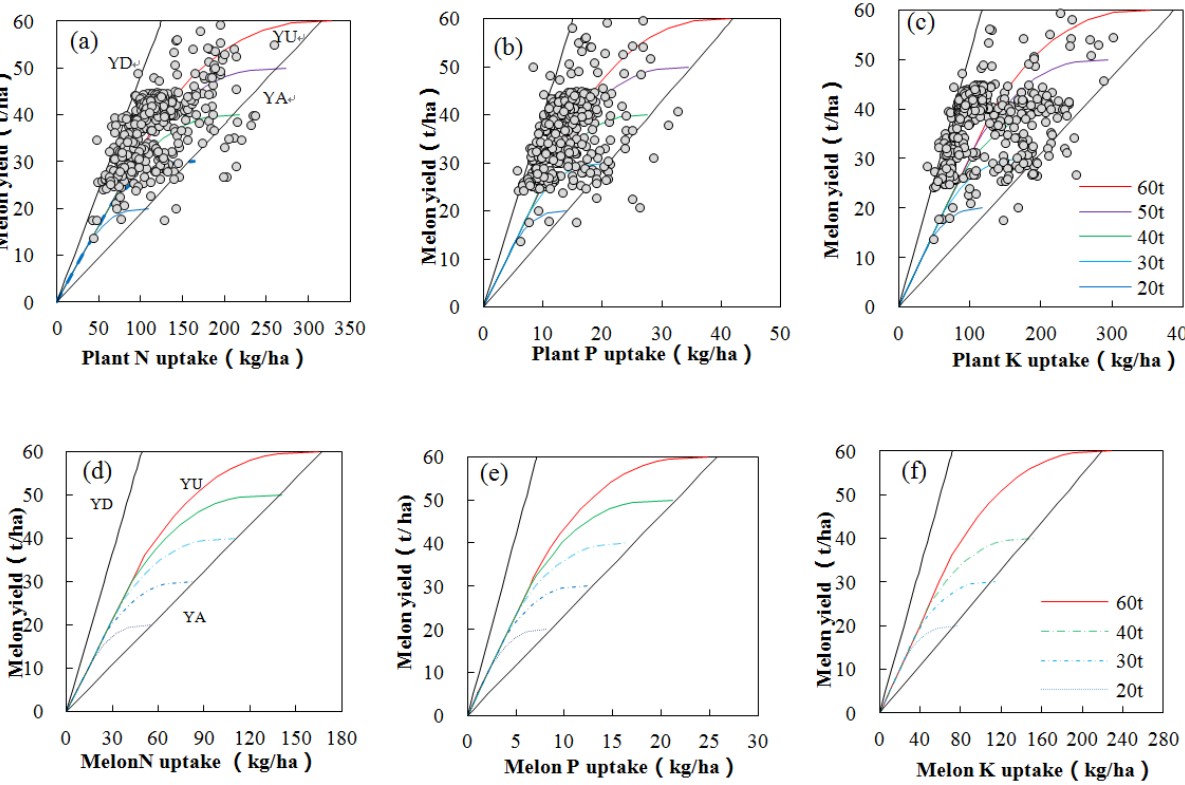

**Figure 2.** Relationship between melon fresh yield and N, P and K accumulation in the total plant above-ground parts at maturity (**a–c**) and fleshy melon N, P and K dry matter (**d–f**) under different target yields stimulated by the QUEFTS model. YD, YA and YU represented maximum dilution, maximum accumulation and balanced uptake of N, P and K in above-ground dry matter or in the fleshy fruit dry matter.

**Table 6.** Balanced nutrient uptake requirements, internal efficiencies (IE, kg fleshy fruit per kg nutrient) and reciprocal internal efficiencies (RIE, kg nutrient per 1000 kg fleshy fruit) of N, P and K simulated by the QUEFTS model to achieve specific melon yield targets and a potential yield of 60 t/ha.

| Melon Yield (t/ha) | Nutrient Uptake of Above-Ground (kg/ha) | | | Internal Efficiency (kg/kg) | | | Nutrient Uptake of Fruit (kg/ha) | | | Ration in Flesh Fruit (%) | | |
|---|---|---|---|---|---|---|---|---|---|---|---|---|
| | N | P | K | N | P | K | N | P | K | N | P | K |
| 4 | 11.6 | 1.53 | 12.9 | 345.3 | 2612.6 | 310.1 | 5.4 | 0.9 | 7.5 | 47.0 | 59.5 | 58.2 |
| 8 | 23.2 | 3.06 | 25.8 | 345.37 | 2612.6 | 310.1 | 10.9 | 1.8 | 15.0 | 47.0 | 59.5 | 58.2 |
| 12 | 34.8 | 4.59 | 38.7 | 345.3 | 2612.6 | 310.1 | 16.3 | 2.7 | 22.5 | 47.0 | 59.5 | 58.2 |
| 16 | 46.3 | 6.1 | 51.6 | 345.3 | 2612.6 | 310.1 | 21.8 | 3.6 | 30.0 | 47.0 | 59.5 | 58.2 |
| 20 | 57.9 | 7.7 | 64.5 | 345.3 | 2612.6 | 310.1 | 27.2 | 4.6 | 37.5 | 47.0 | 59.5 | 58.2 |
| 24 | 69.5 | 9.2 | 77.4 | 345.3 | 2612.6 | 310.1 | 32.6 | 5.5 | 45.0 | 47.0 | 59.5 | 58.2 |
| 28 | 81.1 | 10.7 | 90.3 | 345.3 | 2612.6 | 310.1 | 38.1 | 6.4 | 52.5 | 47.0 | 59.5 | 58.2 |
| 32 | 92.7 | 12.2 | 103.2 | 345.3 | 2612.6 | 310.1 | 43.5 | 7.3 | 60.0 | 47.0 | 59.5 | 58.2 |
| 36 | 104.2 | 13.8 | 116.1 | 345.3 | 2612.6 | 310.1 | 49.1 | 8.2 | 67.8 | 47.1 | 59.7 | 58.4 |
| 40 | 116.6 | 15.4 | 129.8 | 343.1 | 2595.7 | 308.1 | 56.2 | 9.4 | 77.5 | 48.2 | 61.1 | 59.7 |
| 44 | 133.3 | 17.6 | 148.5 | 330.0 | 2496.4 | 296.3 | 64.6 | 10.8 | 89.1 | 48.4 | 61.4 | 60.0 |
| 48 | 152.9 | 20.2 | 170.3 | 313.9 | 2374.8 | 281.9 | 74.3 | 12.4 | 102.4 | 48.6 | 61.5 | 60.1 |
| 52 | 176.6 | 23.3 | 196.6 | 294.5 | 2227.6 | 264.4 | 86.0 | 14.4 | 118.7 | 48.7 | 61.7 | 60.4 |
| 56 | 208.7 | 27.6 | 232.4 | 268.3 | 2029.8 | 240.9 | 102.2 | 17.1 | 140.9 | 49.0 | 62.0 | 60.6 |
| 60 | 313.6 | 41.4 | 349.2 | 191.4 | 1447.6 | 171.8 | 153.5 | 25.7 | 211.7 | 48.9 | 62.0 | 60.6 |

The QUEFTS model predicted that an optimal nutrient requirement of 2.9 kg of N, 0.4 kg of P and 3.2 kg of K would be needed to produce 1000 kg/ha of melons, with the corresponding IE values being 345.3 kg/kg, 2612.6 kg/kg and 310.0 kg/kg for N, P and K, respectively, while the corresponding ratio of the estimated data in the linear part was 7.3:1:8 (Table 6). The average IEs of N, P and K obtained from field experiments were 306.9, 2462.5 and 299.1 kg/kg, respectively. While the corresponding ratio of nutrient uptake from field experiments in the melon database was 8.0:1:8.2. For comparison, the lower IE of N, P and K in under field conditions could be due to nutrient imbalances or differences in yield potential of various experimental sites. Additionally, smallholder farming systems are also characterized by a multitude of non-nutrient yield limiting factors including pests, diseases, micro-climatic variability and management factors. The IE values derived from the slope for only the linear portion of the predicted relationship curve and RIE predicted using QUEFTS decreased when the target yield reached above 60–70% of the potential yield [28]. Our investigations showed that this model could be used to determine the fertilizer application rate and that some of the N and P uptake data were distributed near maximum accumulation and maximum dilution levels and not close to the optimal nutrient uptake curve. This may be due to an abnormal relationship between nutrient uptake and yield as well as differences in the yield potential of the experimental sites. It was observed that excessive amounts of N and P fertilizers were applied in some areas, whereas the application was insufficient in other areas of melon production (Figure 2a,b). Furthermore, a small proportion of data was concentrated near the lower levels, which reflects an excessive K uptake by the plants (Figure 2c), likely due to the presence of large K reserves in the soils of the main melon-producing areas. Moreover, it was observed that disturbances in the normal growth process owing to the excess or lack of nutrient supplementation occurred either during or after the fruiting stage. These results confirm the importance of examining the nutrient requirements of a crop in relation to yield potential and demonstrate that greater attention should be paid to apply the appropriate nutrients during the fruiting period when proposing fertilizer recommendations [16].

In order to prevent nutrient depletion of soil and maintain soil fertility, nutrients that are removed by fruit and above-ground plant dry matter must be returned to the soil [45]. The nutrient removal amount of melon fruit, estimated by the QUEFTS model can be used to adopt a rational approach for fertilizer application [18]. The constants *a* and *d* for the fresh fruit nutrient uptake were calculated after eliminating the upper and lower 2.5th percentiles of all fleshy fruit nutrient IE values. The results showed that the curves of fleshy fruit nutrient removal were consistent with those of the total above-ground nutrient absorption for the potential yield of 20–60 t/ha (Figure 2d–f). Analysis of the model indicated that 1.4, 0.2 and 1.9 kg of N, P and K, respectively, was required to compensate for nutrients removed by the production of 1000 kg of melon, and the N:P:K ratio in the fruit was 7:1:10. Compared with the nutrient absorption in the total above-ground plant, approximately 47.0%, 59.5% and 58.2% of N, P and K, respectively, were accumulated in the fleshy fruit and removed from the field during harvesting (Table 6). This result indicates that P and K were needed in high quantities to produce 1000 kg of fruit; hence, these values must be considered while preparing fertilizer recommendations for melon as well as to maintain soil fertility.

### 3.5. Field Validation of the QUEFTS Model

Multiple field experiments using the NE system were conducted in 2018 and 2020 in the northwest regions of China to validate and calibrate the QUEFTS model. The observed data used for validation were obtained from field experiments in which the rates of N, P and K application were recommended by the NE system, based on SSNM and applied for the entire growing period and the QUEFTS model. RMSE and n-RMSE values were used to evaluate the applicability of the model. The observed and predicted values were distributed around the 1:1 line (Figure 3), and the *p*-values for N, P and K were 0.7496, 0.6791 and 0.7321, respectively. The RMSE values of N, P and K were 18.2, 2.9 and 27.9 kg/ha,

respectively, while the corresponding n-RMSE values of N, P and K were 9.6%, 12.9% and 18.5%, respectively. These lower values for RMSE and n-RMSE indicated that the data of nutrient absorption levels simulated by the QUEFTS model agreed well with observed nutrient absorption levels and that there was no significant difference between observed and simulated nutrient absorption. Additionally, for Xinjiang, some values deviated from the 1:1 line, indicating excess K uptake owing to the greater initial K availability in the soil (Table 1). However, for Inner Mongolia and Ningxia, only one point was far from the 1:1 line, indicating that there was no clear luxury K uptake in of these areas mainly because the initial K availability in soil was much lower than that in the Xinjiang and Gansu sites. The findings of the field experiments suggested that nutrient uptake simulated by the QUEFTS model can be applied to develop fertilizer recommendations for melon and help optimize nutrient management practices and decrease nutrient loss.

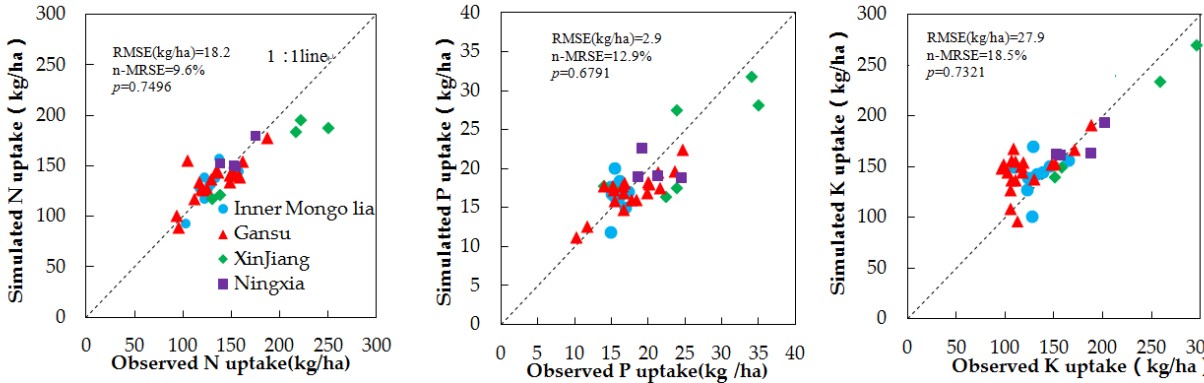

**Figure 3.** Nutrient requirements of N, P and K in total above-ground parts and fruit and removal ratio at different target yields simulated by the QUEFTS model for melon in China.

## 4. Conclusions

Our investigations on the relationship between melon nutrient uptake requirements and yield were based on the evaluation of fertilizer response trials. In this study, large-scale datasets collected from different melon growing areas were used to evaluate optimal nutrient requirements, and the constants of *a* and *d* were used for running the QUEFTS model to estimate balanced nutrient requirements. It also predicted that 2.9, 0.4 and 3.2 kg/ha of N, P and K were required to produce 1000 kg of above-ground parts of melon, while the corresponding IE values were 345.3, 2612.6 and 310.0 kg/kg for N, P and K, respectively. According to the QUEFTS model 1.4 kg of N, 0.2 kg of P and 1.9 kg of K will be removed by harvesting the fleshy fruit, accounting for 47.0%, 59.5% and 58.2% of N, P and K, respectively, of the total above-ground parts. Multiple-site field validation experiments indicated that the QUEFTS model could be used to estimate balanced nutrient requirements, with the estimated values having a good correlation with the observed nutrient uptake values. These findings indicated that the QUEFTS model could be used to support the nutrient expert system for guiding melon nutrient management, which would be a feasible nutrient management strategy to optimize melon yield and improve nutrient use efficiency.

**Author Contributions:** M.W., L.H., S.Y., C.W. collected and analyzed the data, conducted research work and prepared the manuscript; X.X., P.H., Y.Z. and W.Z. designed and supervised the study; M.W., L.H., S.Y. and C.W. revised and approved this manuscript for publication. All authors have read and agreed to the published version of the manuscript.

**Funding:** This research was funded by the National Key Research & Development Program of China (2016YFD0200104) and Youth Science and Technology Fund Project of Gansu provincial of China (21JR7RA724).

**Conflicts of Interest:** The authors declare no conflict of interest.

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
