# Peer review of "Estimating Nutrient Uptake Requirements for Melon Based on the QUEFTS Model"

_agronomy, doi:10.3390/agronomy12010207_

Round 1

Reviewer 1 Report

Very nice work..

  • In the introduction, I suggest comparing the comparison of melon production in China with other countries with a similar climate and/or soil;
  • Results and discussion - Line (215 and 216) - This yield was greater than average yield in other parts of the world at 27.3t/ha [33] - what parts? To specify. 

Reviewer 2 Report

The authors determined the relationship between melon yield and nutrient uptake across a wide range of melon producing environments in China, and estimated melon N, P, and K requirements using the QUEFTS model. Finally, they validated the nutrient uptake requirements through on-farm experiments. The research work is of significant interest to the readers and will help develop fertilizer recommendations for improving melon yield and nutrient use efficiency. Furthermore, the science behind the study is sound. Therefore, I am in favour of accepting this manuscript in Agronomy. However, some minor issues should be addressed before being accepted for publication in this Journal.

L16 fertilizer. Replace ‘fertiliser’ with ‘fertilizer’ throughout the manuscript.

L27 above-ground. Be consistent

There are too many misspacing between words. Please check the entire manuscript and correct them.

L43 ‘semi-arid’, be consistent in the use

L56 Use “knowing” instead of “having knowledge of”

L107 melon-producing

L126, Table 1, Soil characteristics in main……………………

L137 5th… 7th

L170 the excessive

L201 simulated??

L342 parametrized

L464 Conclusion is unnecessarily stretched. Make it compact by summarizing the key results.
